# A Guide for Using Transmission Electron Microscopy for Studying the Radiosensitizing Effects of Gold Nanoparticles In Vitro

**DOI:** 10.3390/nano11040859

**Published:** 2021-03-27

**Authors:** Ioanna Tremi, Sophia Havaki, Sofia Georgitsopoulou, Nefeli Lagopati, Vasilios Georgakilas, Vassilis G. Gorgoulis, Alexandros G. Georgakilas

**Affiliations:** 1DNA Damage Laboratory, Department of Physics, School of Applied Mathematical and Physical Sciences, Zografou Campus, National Technical University of Athens (NTUA), 15780 Athens, Greece; ioannatremi@mail.ntua.gr; 2Molecular Carcinogenesis Group, Department of Histology and Embryology, School of Medicine, National and Kapodistrian University of Athens, 75 Mikras Asias Street, 11527 Athens, Greece; shavaki@med.uoa.gr (S.H.); nlagopati@med.uoa.gr (N.L.); vgorg@med.uoa.gr (V.G.G.); 3Department of Material Science, University of Patras, 26504 Patras, Greece; up1057156@upatras.gr (S.G.); viegeorgaki@upatras.gr (V.G.); 4Biomedical Research Foundation, Academy of Athens, 4 Soranou Ephessiou Street, 11527 Athens, Greece; 5Faculty Institute for Cancer Sciences, Manchester Academic Health Sciences Centre, University of Manchester, Manchester MP13 9PL, UK; 6Center for New Biotechnologies and Precision Medicine, Medical School, National and Kapodistrian University of Athens, 75 Mikras Asias Street, 11527 Athens, Greece

**Keywords:** gold nanoparticles, transmission electron microscopy (TEM), radiosensitization, immunocytochemistry, silver-enhancement, immunogold-labelling

## Abstract

The combined effects of ionizing radiation (IR) with high-z metallic nanoparticles (NPs) such as gold has developed a growing interest over the recent years. It is currently accepted that radiosensitization is not only attributed to physical effects but also to underlying chemical and biological mechanisms’ contributions. Low- and high-linear energy transfer (LET) IRs produce DNA damage of different structural types. The combination of IR with gold nanoparticles may increase the clustering of energy deposition events in the vicinity of the NPs due to the production mainly of photoelectrons and Auger electrons. Biological lesions of such origin for example on DNA are more difficult to be repaired compared to isolated lesions and can augment IR’s detrimental effects as shown by numerous studies. Transmission electron microscopy (TEM) offers a unique opportunity to study the complexity of these effects on a very detailed cellular level, in terms of structure, including nanoparticle uptake and damage. Cellular uptake and nanoparticle distribution inside the cell are crucial in order to contribute to an optimal dose enhancement effect. TEM is mostly used to observe the cellular localization of nanoparticles. However, it can also provide valuable insights on the NPs’ radiosensitization pathways, by studying the biochemical mechanisms through immunogold-labelling of antigenic sites at ultrastructural level under high resolution and magnification. Here, our goal is to describe the possibilities, methodologies and proper use of TEM in the interest of studying NPs-based radiosensitization mechanisms.

## 1. Introduction

IR is widely used both for therapeutic (such as in cancer) and for diagnostic purposes. Despite the advantages in cancer treatment exposure to IR is often associated with adverse health effects [1]. This is due to the fact that radiation not only damage cancer tissue but also healthy tissue. Especially in severe cases of cancer, radiation therapy is efficient only at high doses of radiation resulting in increased toxicity to the surrounding tissue. In recent years, the idea of using radiosensitizers to enhance radiation effect in tumor therapy, while minimizing the damage to the normal tissue, has gained a lot of attention [2,3]. Metal nanoparticles (NPs) such as gold (high atomic number Z = 79) can increase radiosensitivity by increasing local energy deposition by using differential absorption coefficient of high atomic number material, compared to the soft tissue [4]. In this paper we used colloidal solutions of two types of gold nanoparticles (GNPs), citrate-capped GNPs and PEG-capped GNPs. It should be noted that, colloidal stability of nanoparticles especially when they are dispersed in biological media is crucial in order to preserve their correct use as aggregation leads to different biological responses [5]. The main factors of colloidal stability are pH and ionic strength. The physicochemical properties of the NPs are predefined by their structure and material but are also highly affected by their interaction with the environment in terms of temperature, pH, salt, proteins and different cells [6]. Several compounds are utilized to lower the surface tension (interfacial tension) between a liquid and a solid, between two liquids or between a gas and a liquid, acting as detergents, wetting agents, emulsifiers, foaming agents, or dispersants [7]. The surface coating chemistry since it is the first part of the nanoparticle that comes in contact with the biological environment, it does not only provide biocompatibility but also it defines their colloidal stability and their fate (uptake, excretion, degradation). Various coatings have been developed to improve the stability and biocompatibility of GNPs such as polymers (e.g., polyethylene glycol (PEG)), thiols, citrate, peptides, lipids and other surfactants or inorganic coatings such as silica [8]. Many of these coatings for example charged ligands (e.g., citrate) or other surfactants do not stabilize the NPs sufficiently especially in the presence of biological media, salts or protein containing media. On the other hand, polymeric coatings (e.g., PEG) provide NPs that are stable in buffer solutions or cell culture media (containing free proteins or not) [9]. Finally, protein-coated NPs also exhibit high colloidal stability in biological environments, in the presence of salt and of other proteins and are highly stable even inside biological media. It should be noted though that since proteins are sensitive to environmental pH, the stability of protein-coated NPs is also pH dependent [9].

The radiosensitizing effects of GNPs are attributed to physical, chemical and biological mechanisms of actions and researchers now point out that it is a synergy of events that leads to increased radiosensitization [10]. Biological mechanisms of actions, include oxidative stress due to Reactive Oxygen Species (ROS) production, DNA damage, cell cycle effects and apoptosis. The most commonly used experimental techniques to study the effects of radiation in combination with GNPs radiosensitization include clonogenic survival assays (e.g., for apoptosis), flow cytometry (e.g., for cell cycle effects), MTT assays (e.g., toxicity), immunofluorescence (e.g., DNA damage and other types of damage) and/or gel electrophoresis [11,12,13]. Transmission Electron Microscopy (TEM) has not been used extensively to study NPs radiosensitization until now. Its use is limited on studying only the cellular uptake and localization of NPs inside the cell, including gold [14,15,16]. However, TEM has been already used to study and quantify radiation damage (both for low- and high-LET) and more explicitly DNA damage [17,18].

TEM is an important tool used in biomedical research, to investigate the ultrastructure of tissues, cells, organelles and macromolecular complexes. At a cellular and subcellular level, radiation response may manifest in minor reversible or irreversible morphological changes, depending on radiation dose, type of radiation as well as cellular type. These changes generally refer to either the cellular shape itself or to other organelles such as mitochondria, cell membrane, autophagosomes, lysosomes, cytoskeleton, endoplasmic reticulum and Golgi complex [19]. More precisely, changes in the number and size of microvilli, retraction of pseudopods, or rounding up of flatten cells (e.g., endothelial cells, human fibroblasts) may be observed following radiation [20,21]. Concerning the cytoskeleton, disorganization of actin network and disruption of filaments have been observed in several types of cells exposed to IR, in some cases in doses ranging from 0.5 Gy–1 Gy [21,22,23,24]. Many studies also have reported changes in mitochondria, such as elongation, branching and increase in size or disruption of inner and outer membranes [25,26,27]. Golgi fragmentation or rearrangement as well as changes in the number and size of lysosomes and autophagic vacuoles are also present after exposure to IR [19,26,28]. Due to all of these post- IR effects it would be very beneficial to use electron microscopy to study these morphological changes after IR with the presence and absence of GNPs. Electron microscopy provides high resolution images resulting from the use of electrons as the source of illuminating radiation. TEM is employed, observing thin sections (60–90 nm thick) of tissues or cells in conjunction with the application of a variety of techniques, such as immunogold-labelling and negative staining, to answer specific questions. It offers the most powerful magnification and is able to yield information on cellular features, such as shape, size and structure [29]. In addition to receiving ultrastructure information and visualize NPs cellular uptake and localization, it enables accurate detection of different gold-labelled markers at a single-molecule level (e.g., DNA repair factors within the chromatin ultrastructure) [30,31,32]. Nonetheless, when considering using TEM one has to appreciate its limitations and disadvantages. The main limitations are that the required equipment is expensive, the applied techniques are time-consuming and the laboratory personnel must be appropriately trained.

Here we aim to introduce TEM as an efficient way to study the biochemical effects of gold nanoparticles after ionizing radiation in vitro, by summarizing the most important experimental procedures accompanying TEM, along with their protocols. These include, studying the cellular uptake of NPs, as well as identifying specific markers indicating damage or cellular stress (e.g., DNA damage, oxidative stress, cellular senescence, autophagy and apoptosis). Specific markers can be identified in vitro through TEM via immunocytochemistry applying immunogold-labelling [1,2]. This method uses colloidal solutions of gold-conjugated antibodies as secondary antibodies, which localize the antigenic sites under study. As an example of elaborate adaptation and improved use of TEM and to represent how this method can be used to study GNPs radiosensitization, we performed single and double immunogold-labelling in PC3 (prostate cancer) cells pre-incubated with citrate-capped GNPs. By this way, we achieved to monitor the formation and repair of isolated and clustered DNA damage (DSBs and base lesions) induced by X-rays. Concerning the cellular uptake of GNPs, in this work, we used 15 nm of both citrate-capped and PEG-capped GNPs. At the same time, we introduce the silver enhancement technique as a way to enhance the signal and visualize easier smaller sizes (5 nm or less) of GNPs. Small and ultrasmall GNPs are a promising mean to enhance radiation damage, compared to large ones. Subsequently, it is of great importance to monitor their cellular uptake and their localization inside the cell environment, before irradiation. Silver enhancement is usually performed after immunogold staining to help visualize smaller (nano-) gold particles [3]. During silver-enhancement, the colloidal gold serves as a nucleation site for the deposition of metallic silver. The silver encapsulation increases the size of gold, imparting in a black colored particle to the stained background. In this work, we have used this technique on thin sections of PC3 cells pre-incubated with 5 nm gold nanoparticles (colloidal gold) in order to enhance the localization signal. In total, in this technical note we present and summarize all the experimental and methodological technicalities that one has to follow in order to study the radiosensitizing effects of gold nanoparticles through TEM.

## 2. Methods and Results

### 2.1. Cell Culture

PC3 (prostate cancer) cells were cultured in 100 mm petri dishes at 37 °C in an atmosphere with 5% CO_2_. Cells were grown in Dulbecco’s modified MEM (DMEM) medium supplemented with 10% fetal bovine serum (FBS), 1% L-glutamine and antibiotics. Exponentially growing cells were incubated with gold nanoparticles after reaching a ~50% confluency.

### 2.2. Cell Incubation with Gold Nanoparticles

Cell incubation time with gold nanoparticles varies. Cells must be incubated with NPs for some hours, so that they can internalize them. Adherent cells are usually plated on a petri dish and after cells are attached to the substrate old medium is discarded and replaced by fresh growth medium containing NPs. Nanoparticles may be in powder form or in liquid stock suspension such as colloidal gold. Nanoparticles are then resuspended in fresh growth medium in order to reach the final desired concentration. Incubation time with NPs varies from 2–4 h up to 24 h. After a few hours (e.g., 7–10 h) the uptake rate reaches plateau [4]. However, most researchers studying gold nanoparticles radiosensitization incubate cells with NPs for 24 h, leading to a significant increased amount of internalized particles [5,6]. Longer incubation times are rarely used, since after 48 h–72 h NPs are starting to be excreted by the cell. In the present study, cells were incubated with gold nanoparticles (PEG capped GNPs and citrate capped GNPs) for 24 h. Afterwards, cells were washed thoroughly, at least three times with PBS to remove NPs not internalized by the cells. Then, cells were incubated with fresh growth medium and some of them were treated according to irradiation procedure (see Section 2.3). The irradiated or not PC3 cells after their incubation with GNPs were fixed and further treated for TEM processing to monitor cellular uptake, as described below.

### 2.3. Irradiation of the Cells

Cells, after their incubation with nanoparticles, were washed thoroughly with phosphate buffer saline (PBS) and incubated in fresh growth medium. As a means to combine GNPs with low radiotherapeutic doses, cells were then irradiated with 1 Gy using an X-ray machine (GE-Healthcare) operated at 320 kV, 10 mA with a 1.65 mm Al filter (effective photon energy approximately 90 kV), at a distance of 50 cm and a dose rate of approximately 1.3 Gy/min [7,8]. Cells were then returned to the incubator at 37 °C immediately after exposure to IR for 30 min before their further fixation and treatment according to the chosen TEM procedure to study nanoparticle radiosensitization.

### 2.4. Cell Fixation and Preparation for TEM

Because cultured cells may be adherent or non-adherent (in suspension), detailed handling methods may vary. However, the basic steps for TEM specimen preparation are the same: fixation in aldehyde buffer solution, embedding in gelatin, post fixation in osmium tetroxide, dehydration in ethanol and embedding in resin. Chosen chemical fixative and buffer solutions depend on the aim of the ultrastructural study of the cells using TEM. Glutaraldehyde develops intra- and intermolecular bonds between protein molecules and other macromolecules (e.g., carbohydrates) making the structures permanent irreversibly. Even though formaldehyde penetrates cells faster than glutaraldehyde, it doesn’t create so many bonds with the protein molecules and also the fixation process might be reversible. In order to study morphologically the cellular uptake of GNPs, a 2.5% glutaraldehyde solution in 0.01 M PBS for 30 min is used. For immunocytochemical applications, a solution of mixture of 3% paraformaldehyde and 0.5% glutaraldehyde in 0.1 M PB (sodium phosphate buffer) can be used as fixative, to better preserve the antigenicity and the ultrastructure of the cells simultaneously. Once fixed, cultured cells (adherent or not) are centrifuged and embedded in gelatin to facilitate their handling as tissue fragments.

#### 2.4.1. Embedding in Gelatin of Cells Attached to a Substrate (Adherent Cells)

Cells attached to a substrate, such as a petri dish, are chemically fixed and then collected in an Eppendorf tube for embedding in gelatin solution. Specifically, growth medium is replaced by 2.5% glutaraldehyde solution in PBS, pH 7.2–7.4, for 30 min at room temperature (RT). After fixation is completed, cells are washed thoroughly with PBS, are then harvested using scraper and finally are collected into an Eppendorf tube. Cells are then centrifuged at 800× *g* for 5 min at RT. The supernatant is aspirated and the cells are resuspended in 4% gelatin aqueous warmed solution. The cells were further centrifuged at 800× *g* for 5 min at RT and the Eppendorf tubes with the formed cell pellets embedded in gelatin are cooled on ice, so that the gelatin become solidified. At least one million of cells is necessary to form a visible cell pellet. Gelatin containing the cell pellet is then cut into small cubes of ~1 mm^3^ volume under a stereoscope, using a sharp razor blade. These small pieces are then immersed in PBS at 4 °C until ready for processing. Cell pellets containing gold nanoparticles usually look darker compared to the nanoparticle-free pellets that are whiter. This is because of the different materials used for nanoparticle construction (e.g., citrate or titanium oxide).

#### 2.4.2. Cells in Suspension (Lymphocytes)

Usually those who study in situ nanoparticles’ radiosensitization may use non-adherent cells. Since NPs are not attached to a substrate, this may create controversies as to whether or not NPs penetrate cells without immediate cell-nanoparticle contact. Lymphocytes are very useful cellular model in studying NPs radiosensitization since they are in G0 phase of the cell cycle. These cells can also reenter the growth cycle in response to specific stimulation by mitogens. Cell cycle phase is crucial when studying the effects of ionizing radiation with the presence or absence of GNPs. Radiosensitivity fluctuates between phases because of chromatin condensation or decondensation and because each cell cycle phase follows different DNA repair pathways [9,10]. One of the main biological mechanisms of action of GNPs is the production of ROS which may lead to cell cycle arrest. Lymphocytes can either be used directly in G0 phase or they can be subcultured and reenter the cell cycle as synchronized cells. At first, peripheral blood from healthy individuals is drawn in heparinized tubes. Human lymphocytes can be directly separated from heparinized blood samples using Biocoll separating solution (Biochrom). The blood sample is diluted 1:2 in cell culture medium (RPMI-1640) without FBS, and is carefully layered on top of an equal amount of Biocoll in a test tube before centrifugation at 400× *g* for 20 min. Collected lymphocytes (middle layer) are then washed with 10 mL culture medium, centrifuged at 300× *g* for 10 min and kept in petri dish or flask containing culture medium (RPMI-1640 supplemented with 10% FBS, 1% glutamine and antibiotics). These cells can then be incubated with GNPs for 24 h. Alternatively, cultures are set up by adding 0.5 mL of whole blood to 5 mL of RPMI-1640 medium supplemented with 10% FBS, 1% phytohemagglutinin (PHA), 1% glutamine and 1% antibiotics (penicillin, streptomycin). Cultures must then be incubated at 37 °C in a humidified chamber with 5% CO_2_ for 72 h. Cells can then again be incubated with GNPs for 24 h. Finally, lymphocytes can be fixed as described above as a suspension or as a cell pellet. After several washes lymphocytes are embedded in gelatin and cut into small cubes under a stereoscope. At least a million cells (≥10^6^) is necessary to create an adequate visible pellet for TEM applications.

### 2.5. Processing and Embedding in Epoxy Resin

This standard process of preparation and embedding of specimens in epoxy resin is applied to samples for studying morphologically the gold nanoparticle cellular uptake. After chemical fixation with aldehydes, specimens (cells-gelatin fragments) are subsequently treated with 1% aqueous osmium tetroxide (OsO_4_) solution for 1 h at 4 °C. Therefore, fixation is a two-fold procedure, with glutaraldehyde preserving the cellular ultrastructure by protein crosslinking and osmium tetroxide retaining the lipids in the cell membranes, that otherwise would be extracted during later steps. Specifically, osmium tetroxide is a strong oxidizer that reacts with the double bonds of unsaturated fatty acids and with proteins. As a heavy metal osmium acts also as a pigment and enhances the contrast when viewing specimens in TEM. However, post-fixation with OsO_4_ covers most of the antigenic sites, reducing consequently the antigenicity. For that reason this step should be skipped, if immunocytochemistry is to be performed. Nevertheless, if it is already used, strong antioxidants such as hydrogen peroxide (H_2_O_2_) have to be employed to restore antigenicity before moving on. After fixation and post-fixation, cells are gradually dehydrated in organic solvents solutions (usually ethanol or methanol) of increasing concentration (25%–50%–70%–95%–100% ethanol). After the cells reach 100% ethanol, they are infiltrated usually with an epoxy resin which is a plastic monomer. Epoxy resins (e.g., Araldite) are the main class of embedding resin used, as they preserve and protect the ultrastructure of cells, are not so much effected by heat during polymerization, are easily cut into thin sections and provide high stability under electron beam during observation. In the results shown here, a mixture of epoxy resins has been used during infiltration composed of araldite, glycid ether 100, DDSA and DMP-30 (see Table 1). Infiltration comprises of four steps. Due to high viscosity of epoxy resins, infiltration is gradually performed by increasing concentration mixtures of epoxy resin/propylene oxide, starting with propylene oxide alone. Finally, specimens are placed in embedding molds filled with epoxy resin in order to form resin blocks. Each block contains one specimen. Molds are placed at 60 °C for ~24 h, so that the mixture of epoxy resins is polymerized and therefore, the blocks are solidified.

### 2.6. Processing and Embedding in Acrylic Resin

As it is already mentioned TEM can also be used for immunolocalization of specific markers. This can be done for irradiated cells, as well as for irradiated cells pre-incubated with gold nanoparticles. In order to perform immunocytochemistry to locate for example DNA damage markers using TEM it is better to embed cell samples in acrylic resins applying the Progressive Lowering of Temperature (PLT) method, preserving the antigenicity. This method is chosen when the antigenic sites of interest are not so durable and are sensitive to processing and polymerization in high temperatures. Cells are mild fixed with a mixture of aldehydes (glutaraldehyde-formaldehyde) as mentioned in the fixation section and then are dehydrated in solution of organic solvent (usually ethanol) of increasing concentration. The basic principle of PLT is the progressive lowering of temperature (below 0 °C) as the concentration of the organic solvent gradually increases. The entire dehydration, embedding and UV-polymerization procedure is carried out inside the chamber of specific instrument (e.g., EM AFS, Leica Instruments) in very low temperatures achieved by liquid nitrogen [11]. This, better ensures the protection of the antigenic sites, prevents the dissolution of cellular components and reduces any changes that may occur in the structure of proteins. The protocol described here refers to Lowicryl HM20 resin (see Table 2), which is nonpolar hydrophobic and is polymerized at −50 °C [11]. Therefore, in acrylic resins the immunocytochemical reaction is enhanced compared to epoxy resins [12]. Moving on to the experimental procedure (which in this protocol lasts five days), after fixation cell samples are placed inside the chamber of the LEICA EM AFS and are dehydrated in increasing concentrations (30%–50%–70%–100%) of anhydrous alcohol (ethanol) by lowering the temperature progressively from 0 °C to −50 °C. Time duration of each dehydration step is specific while also some intermediate pauses exist for transitional lowering of temperature. To be noted, it is of crucial importance to prevent samples from freezing. Therefore, care has to be taken to keep the temperature of the dehydration mixture above its freezing-point and to have precise preparation. That is why dehydrated agents (e.g., molecular sieve) must be used for all the organic solvents. The lowest temperature (−50 °C) is applied in the last step of dehydration (100% anhydrous ethanol). After dehydration, the infiltration begins using acrylic resins in different concentration ratios. In the first steps of infiltration mixtures of Lowicryl HM20 resin and ethanol at −50 °C are used and then follows the infiltration with 100% Lowicryl HM20 at −50 °C overnight. The next day, the resin is renewed and cell samples are embedded in resin inside Leica capsules and polymerized under a UV lamp (wavelength: 350 nm, power: 6 W) at −50 °C for 48 h and at 0 °C for 24 h. It should be mentioned that osmium tetroxide is not used as a complementary fixative in PLT protocol, because it interferes with the penetration of UV light into the specimens resulting in incomplete polymerization. This, however, provides an additional benefit regarding the antigenicity preservation of the tissue.

### 2.7. Sectioning of Resin-Embedded Tissue Blocks

Thin (70–90 nm thick) and semi-thin sections (1 μm thick) of resin-embedded tissues are cut using an ultramicrotome. Acrylic resin blocks are harder than epoxy resin blocks.

#### 2.7.1. Semi-Thin Sections

At first, semi-thin sections have to be obtained in order to observe the specimen under light microscope. So, semi-thin sections have a guiding role providing preliminary information regarding cellular morphology and several technical issues, such as fixation quality and the infiltration adequacy of embedding medium. They also help to decide which block could be selected and cut later into thin-sections for each experimental condition based on the best field of view. Semi-thin sections are cut out from resin-block using glass knife with a mounted homemade tape-trough or plastic trough (commercially available) filled with dH_2_O to collect the fragile sections of ~1 μm thickness, floating on water surface. Sections are then transferred via a metallic loop on a microscopy glass-slide, and dried on a hot plate (70 °C–90 °C). 

##### Staining of Semi-Thin Sections

Dried semi-thin sections are stained with 1% Toluidin Blue O and 1% sodium borate aqueous solution, which is made up of a mixture of equal drops of 2% Toluidin Blue O and 2% sodium borate aqueous solutions, which are first filtered and then mixed just before use. The semi-thin sections are stained approximately for 1 min and are observed under a light microscope.

#### 2.7.2. Thin Sections

Thin sections for TEM observation (~80 nm thick) are usually cut using diamond knife with permanently mounted trough filled with dH_2_O or in some cases with glass knife of high quality. In short, before sectioning, the block edges have to be freshly trimmed with a sharp and clean one-edge razor blade. Thin sections are cut and float on the water surface of the trough. Thin sections are collected and mounted on the grid, lowering it carefully over the sections. However, the use of commercially available fine loop is recommended for section collection to avoid several problems, such as overlapping or damage of the fragile thin sections. For observation of nanoparticles’ cellular uptake under TEM, copper grids of 200 mesh are used for morphologically study, while nickel grids of 200 mesh are selected for immunocytochemically study.

##### Staining of Thin Sections

Thin sections are stained with 7.5% alcoholic uranyl acetate [13] and 0.4% lead citrate [14] before observation. The staining procedure is carried out by floating the grid-mounted sections (section-containing side facing down) on drops of filtered staining solutions in an environment protected from dust. For that reason, a glass petri dish is used for each staining solution. Specifically, a piece of parafilm is placed inside the glass petri dish and drops of uranyl acetate are put in equal numbers with the grids stained with this solution for 25 min, at RT in the dark. Grids are then rinsed with dH_2_O. In another petri dish, parafilm is placed and drops of lead citrate are put in equal numbers with the grids stained with this solution for 3 min. This time, parafilm has to be surrounded with 10 N NaOH solution to avoid the formation of lead carbonate precipitate after the reaction of lead stain with atmospheric CO_2_, leading to image artifacts. Grids are then rinsed quickly with 0.02 N NaOH following dH_2_O. After each rinsing with dH_2_O, excess liquid is wiped off using optical lens tissues and grids are left to dry out and stored in a gridbox.

### 2.8. Transmission Electron Microscopy and Immunocytochemisty

The application of immunocytochemical methods in Transmission Electron Microscopy can be a valuable experimental approach in studying nanoparticle radiosensitization. These methods can lead to the detailed localization of biomolecules in the micro-environment of the cell providing an analysis of high resolution that could not be achieved by any other approach. In order to obtain high-quality images of high resolution, Transmission Electron Microscope should operate at least at 80 kV accelerating voltage with an objective aperture of 30 μm and be equipped with a digital CCD camera, such as Olympus Morada or Olympus MegaView G2, calibrated according to the manufacturer.

Immunocytochemical procedure can be carried out either before or after the embedding of cells in resin (pre-embedding and post-embedding protocols respectively). In any case, the method of choice aims to the best possible preservation of antigenicity and maintenance of ultrastructural cell morphology, and depends on antigen location, antigen sensitivity and antigen quantity. Here, we will present, as the method of choice, the post-embedding immunolocalization method (post-embedding in acrylic resins) to study the nanoparticle radiosensitization effect through the localization of DNA damage markers.

#### 2.8.1. Immunogold Labelling

The ultrastructural detection of antigenic sites in TEM is achieved using markers, which allow the differential scattering of the electron beam. The most common immunohistochemical marker used is colloidal immunogold [15]. Since 1971, when Faulk and Taylor invented the immunogold staining procedure, colloidal gold has been widely used in laboratory settings [16]. The optical (and electron beam) contrast qualities of gold provide excellent detection of antigenic sites with great precision and high resolution. The principle of immunogold labelling in TEM is the use of gold-conjugated antibodies (secondary antibodies) as a method of localizing proteins in cells and tissues. Antibodies conjugated with gold particles of different diameter sizes can be used in double or even triple labelling. The electron-dense gold particles can be easily identified in the transmission electron microscope and enable the quantitation of the gold label in relation to the compartments of interest. Gold labels appear in TEM images as dark particles –like black dots- against a lighter (less electron-dense) background.

#### 2.8.2. Post-Embedding Immunogold-Labelling

Here we will describe the application of the post-embedding immunogold labelling technique. Cells are embedded in Lowicryl HM20 resin according to PLT method, as described in paragraph 2.6. All steps of this immunohistochemical method are performed at room temperature, except for the incubation step with the primary antibodies, which is carried out at 4 °C. Dilutions of primary and secondary antibodies result after standardization procedure in order to have the optimum antibody binding and at the same time the least non-specific signal. After cutting, thin sections of acrylic resins are mounted on formvar-coated Nickel (Ni) grids of 200 mesh. Nickel grids are preferred as they are not eroded by the immunohistochemical reagents. Moreover, formvar-film coating works as a supportive substrate and provides better stability of acrylic resin-sections under electron beam avoiding moving images during TEM observation. Protocol presented here lasts two days. During each step, incubation is carried out by floating the grids on drops of solutions, with the grid side containing the sections facing down. This protocol can be safely performed using Terasaki-well plates (HLA microtest plates) with lid to ensure a clean dust-free incubation environment and proper humidity. Before use, all solutions are filtered using syringe with an adapted filter with cellulose acetate membrane and 0.22 μm pore size. The first day, before incubation with primary antibody, some preparative steps are necessary to prevent non-specific binding incubating the grids on drops of glycine/Tris solution and blocking solution containing normal serum from the donor species and serum albumin (see detailed protocol). Incubation with primary antibody is carried out at 4 °C overnight. Second day, sections are incubated gradually with Tris buffer solutions of increasing pH containing different concentrations of bovine serum albumin (BSA) and Tween-20, in order to enhance specific binding (see reagent preparation for Tris buffer I, Tris buffer II, Tris Buffer III). Grids are then drained and incubated with the gold-conjugated secondary antibody, diluted 1:40, after standardization, to give sufficient specific signal. The protocol used here is described in detailed in the Appendix A.

### 2.9. Use of TEM to Monitor Gold Nanoparticle Cellular Uptake

GNP-induced radiosensitization strongly depends on their cellular uptake, as well as, on their cellular distribution. Nanoparticles are engulfed by cell membrane through specific pathways, but their entry inside the cells is associated with morphology, size and surface charge or coating materials. Most NPs enter the cells through energy-dependent endocytosis pathway. Vesicles that engulf extracellular NPs, are generated from the cellular membrane and are pinched off. The cellular uptake root of NPs plays key role on their radiosensitizing effects and mechanisms of action, as it determines their location and their lifespan inside the cell. For example, some studies show that DNA damage is enhanced when cells, after gold nanoparticles uptake, are irradiated. However, it is acceptable that increased DNA damage can be attributed to NPs when their presence is inside the nucleus or near the perinuclear region, but not further away. Moreover, irradiation of cells incubated with GNPs increase the amount of ROS production. ROS are one of the main byproducts of the interaction of ionizing radiation with water. Increased ROS may be found when GNPs are present, but nanoparticle-radiation induced ROS do not extend far from the surrounding area. Gold nanoparticles usually accumulate only in the cytoplasm and are located inside vesicles, in lysosomes, autophagosomes as well as in the endoplasmic reticulum [17,18,19]. Small single NPs, or NPs with specific coating and/or nuclear targeted surface moieties can be seen also inside the nucleus. Following the procedures and protocols presented here, we can use TEM as a primary basic tool to observe the cellular uptake, cellular localization, as well as any cellular morphological change due to nanoparticle uptake in correlation with or not irradiation. Figure 1 shows representative images of different GNPs (citrate-capped and PEG-capped) cellular uptake in PC3 cells (prostate cancer cell line). Cells presented here, were incubated with 20 ug/mL, 15 nm citrate-capped GNPs (Figure 1a) and 15 nm PEG-capped GNPs (Figure 1b) for 24 h and then fixed in 2.5% glutaldehyde in PBS.

Nanoparticles are known to be distributed heterogeneously throughout the cell environment [20,21] and many times aggregate forming clusters [20,22]. This is also observed here, where NPs are found in groups. Nanoparticle uptake can be also semi-quantified using image analysis software, such as ImageJ, which is an open-access software. All images must be calibrated before use based on the micrographs scale bar. A defined area, such as a specific cellular structure can be selected or drawn and be measured in μm^2^. A binary image has to be also generated by thresholding. Particle count can be done manually on the selected area but also automatically (especially when aggregates are formed), by using ImageJ plugins. Results can be registered as number of NPs per cell, per organelle or as number of NPs per cellular area in μm^2^. For example, obtained data can be presented as histograms to demonstrate the number of internalized GNPs in relation to several studied parameters (e.g., type of nanoparticle, nanoparticles concentration etc.). Figure 2a,b shows representative data, as an example, to semi-quantify gold nanoparticles uptake. Specifically, in each condition, particles were counted from approximately 15 electron micrographs of three different cells at 23,000–36,000× original magnification.

### 2.10. Silver-Enhancement of Gold Nanoparticles

Gold nanoparticles vary in diameter size from 2–5 nm up to 10–60 nm. Nanoparticles ranging from 10–60 nm can be easily distinguished. However, ultrasmall NPs of 1–5 nm diameter size cannot be easily observed. Here we present a method where small GNPs can become visible by silver-enhancement technique. This method was firstly introduced by Danscher and Holgate in 1983 [23,24]. The immunogold silver-enhancement technique is based on the interaction of colloidal gold-conjugated antibodies with silver and is performed to amplify biomolecular signal of ultrasmall particle size gold-conjugated antibodies [25,26,27]. Ultrasmall GNPs (1–5 nm diameter size) were firstly introduced in order to improve the penetration of gold conjugated antibodies into animal cells and tissues and to enhance labelling efficacy [28]. In terms of radiation enhancement, small or ultrasmall GNPs are used often in combination with ionizing radiation, because of their size and physical properties. Several groups have evidenced that small NPs deposit larger doses in their vicinity than bigger ones due to their greater surface to volume ratio [29]. Ultrasmall NPs are often located inside the nucleus, because their smaller size helps them pass through the nuclear membrane pores. This may increase DNA damage of irradiated cells pre-incubated with GNPs significantly, since the accumulation of Low Energy Electrons (LEEs) will result mostly inside the nucleus. It is therefore of great importance to monitor the cellular uptake of smaller gold nanoparticles. Gold NPs are visible though TEM due to their high electron density. However small sizes of NPs (<5 nm) are not easily detected even in very high magnification. For that reason, silver-enhancement technique is often applied to enlarge the size of GNPs and facilitate the detection of small or ultrasmall ones. The reaction between GNPs and silver is specific for the first time period. Gold nanoparticles will nucleate the deposition of dense silver particles which will enlarge after a rapid period of time resulting in easily detectable NPs (see Figure 3). However, silver-enhancement technique is time-dependent and may give a small amount of background signal depending on the incubation time with reagent solutions. Since, after prolonged incubation with silver reagent, silver may be precipitated due to self-nucleation. In the context of experiments monitoring gold nanoparticles cellular uptake, we used 5 nm colloidal gold nanoparticles capped with PEG. PEG enhances biocompatibility and reduces aggregates formation. The incubation time with silver reagents using the nanoprobes HQ silver-enhancement kit is 6 min, which resulted after standardization procedure in order to acquire an adequate amplified signal, with minimum background (see detailed protocol in Appendix A). After silver-enhancement, thin sections were stained with uranyl acetate and lead citrate for TEM observation. Figure 4 shows a representative image of PC3 cells incubated with 5 nm PEG-capped gold nanoparticles and subsequently performing the silver-enhancement technique on thin acrylic-resin sections (post-embedding).

Of note, small GNPs may be visible without silver-enhancement when they are next to each other in groups forming aggregates. However, if the aim is the detection of single NPs, silver-enhancement technique is recommended as a way to enhance single particle signal and become easily visualized. In this paper we introduce the silver-enhancement technique, as a way to monitor the uptake of small GNPs before IR and estimate their localization in the nucleus. This is a very crucial step before radiation, since nanoparticle localization inside the cellular compartments can elucidate the nanoparticle radiosensitization pathways through biochemical mechanisms after IR. However, this method is not proposed for quantification processes, since NPs tend to appear in groups inside the cell or in some cases form aggregates. Silver develops around gold particles, which makes it difficult to distinguish single particles and so quantification may not be ideal.

### 2.11. Immunogold Labelling for DNA Damage Detection

Here we show how immunogold labelling technique in TEM can be also used for immunolocalization of specific antigens after irradiating cells which are already incubated with gold nanoparticles. As an example, we performed immunocytochemistry to detect DNA damage markers with TEM after irradiating cells with GNPs, studying for the first time their radiosensitization, regarding DNA damage, at ultrastructural level. DNA damage and repair are the most basic post IR effects studied in radiobiology. Other biological markers can be also detected inside the cell applying similar procedures using corresponding antibodies. For these representative data, we incubated PC3 cells for 24 h with 100 ug/mL of 15 nm citrate-capped gold nanoparticles. After incubation, cells were irradiated with 1 Gy radiation dose (X-rays). Cells were fixed 0.5 h post IR and were treated according to PLT method, as described in Section 2.6 and then processed for immunocytochemistry purposes as it has been described in Section 2.8. We present this method for both single (Figure 5a) and double immunolocalization (Figure 5b). For single immunolocalization we used γH2Ax to detect double strand breaks and for double immunolocalization we used γH2Ax and OGG1 to detect not only double strand breaks, but also base lesions (such as the 8-oxoguanine) due to oxidative stress from ROS production. Dilutions used for both primary and secondary antibodies were the result of tests aiming at best antigenic binding and at the same time at the least possible non-specific reaction. Thin sections were incubated with both primary antibodies diluted at 1:200 ratio each. Following, sections were incubated with secondary antibodies (gold-conjugated antibodies) diluted at 1:40 ratio. For γH2Ax detection we used a secondary antibody conjugated to 10 nm immunogold particles, whereas for OGG1 detection we used a secondary antibody conjugated to 25 nm immunogold particles. Figure 5a,b represents single and double immunolocalization respectively.

Due to the difference in size, the gold particles are easily distinguished from each other when observed under transmission electron microscope and it is possible to study comparatively the distribution of the two primary antibodies in the cell. In this experiment we used 15 nm gold nanoparticles to radiosensitize cells, which due to coating, size and possibly aggregation (or grouping) inside the cell, do not enter the nucleus but exclusively are localized in the cytoplasm. Since γH2AX and OGG1 signal is expected inside the nucleus and citrate-capped NPs are located in the cytoplasm, it is easy to distinguish the signal between gold particles conjugated to secondary antibodies (10 nm and 25 nm) and citrate-capped gold nanoparticles (15 nm). Therefore, in single immunogold localization of γH2AX, we observed two different gold nanoparticle sizes in the cell (10 nm immunogold for γH2AX in the nucleus and 15 nm GNPs in the cytoplasm), while in double immunogold localization, we observed three sizes of gold particles in the cell (10 nm immunogold for γH2AX and 25 nm immunogold for OGG1 in the nucleus, as well as 15 nm GNPs in the cytoplasm).

In general, when immunogold localization is performed to study the effect of nanoparticle radiosensitization with TEM, it is advisable the size of the NPs used to increase radiosensitivity to be different from the size of the immunogold particles (gold-conjugated antibodies) used to detect the damage. Otherwise, it will be very difficult to identify which nanoparticle corresponds to the nanoparticle used as a radiosensitizer and which one indicates specific damage.

Immunogold signal, in this case representing DNA damage, can be semi-quantified and results can be presented as number of NPs per cell, per organelle or as number of immunogold particles per cellular area in μm^2^. In this case, particles refer to either γH2Ax or OGG1. Quantification analysis can be performed using ImageJ as mentioned before (see Section 2.9). Figure 6 shows representative data in the form of histograms, as an example for this detection method. Specifically, in each condition, particles were counted from approximately 15 electron micrographs of three different nuclei at 28,000–36,000× original magnification.

## 3. Materials

### 3.1. Equipment

Eppendorf benchtop microcentrifuge, chromafil CA-20/25 membrane filters (MACHEREY- NAGEL, cat. no. 729026), clear glass shell vials with snap-caps (Electron Microscopy Sciences, cat. no. 72631-10), color frosted microscopy slides (HDA, cat. no. 7109), copper mesh grids (square, 200 mesh/MERCK, cat. no. G4776), Diatome diamond knife, 2 mm cutting edge, Diatome perfect loop (DZ8, cat. no. 70944), disposable single-side razor blades, Eppendorf tubes (0.5 mL, 1.5 mL, 2 mL), fine tip anti-magnetic Tweezers (Dumont Biologie #5 or #7/MERCK, cat. no. T4537-1EA/T4912-1EA), flat embedding molds (Electron Microscopy Sciences, cat. no. 70900), glass knife maker (Leica EM KMR2), grid storage box (Ted Pella, cat. no. SB100), hot plate (at least 70 °C), hot plate with magnetic stirrer, Leica EM AFS, optical lens tissues (Electron Microscopy Sciences, cat. no. 71700), oven adjusted at 60 °C, Parafilm M, pH meter (Hanna), stereo microscope, syringes (2.5 mL, 10 mL), Terasaki plate, 72 well, with lid (Greiner Bio-One, cat. no. 769190), ultramicrotome (Leica Ultracut R), water bath (adjusted at 40 °C or 60 °C), Whatman paper/MERCK cat. no. WHA1452125).

### 3.2. Reagents

25% glutaraldehyde EM grade (SERVA, cat. no. 23114), absolute ethyl alcohol (Merck, cat. no. 34852-M), acetone (MERCK, cat. no. 34850), araldite (SERVA, cat.no. 13824), bovine serum albumin—BSA (MERCK, cat. no. A7638), chloroform (MERCK, cat. no. 366927), DDSA (EMS, cat. no. 13710), dibutyl phthalate (SERVA, cat. no. 32805), DMP-30 (SERVA, cat. no. 36975), fish gelatin (MERCK, cat. no. G7765), gelatin from porcine skin (MERCK, cat. no. G2500), glycid ether 100 (SERVA, cat. no. 21045), glycine (MERCK, cat. no. G7403), goat normal serum (Agilent), HQ silver-enhancement kit (Nanoprobes, cat. no. 25C814), lead citrate (FERAK BERLIN, cat. no. 81753), liquid nitrogen, lowicryl HM20 Kit (Polysciences, cat. no. 15924), Na2HPO4 H2O, disodium hydrogen phosphate dihydrate (MERCK, cat. no. 1065801000), NaH2PO4 H2O, Sodium dihydrogen phosphate monohydrate (MERCK, cat. no. 1063461000), NaOH (MERCK, cat. no. 106498), osmium tetroxide, 4% solution (Polysciences, cat. no. 0972A-20), paraformaldehyde (MERCK, cat. no. 158127), phosphate buffer saline (PBS) (Gibco, PBS tablets cat. no. 18912-014), primary antibodies: mouse monoclonal anti-gamma H2A.X (phospho S139) (Abcam, cat. no. ab22551), rabbit polyclonal anti-OGG1 (Novusbio, cat. no. NB100-106), propylene oxide (SERVA, 33715), Secondary antibodies/immunogold: goat anti-mouse IgG (Gam)—10 nm (Aurion, cat. no. 810.022), goat anti-rabbit IgG (Gar)—25 nm (Aurion, cat. no. 825.011), Sorenson’s phosphate buffer (PB), toluidine blue O (FERAK BERLIN, cat. no. 52040), tris (SERVA, cat. no. 37180), tween-20 (SERVA. Cat. no. 37470), uranyl acetate (SERVA, cat. no. 77870).

#### Reagent’s Preparation

Sorenson’s Phosphate Buffer (PB): Two solutions 0.2 M Na_2_HPO_4_ 2H_2_O (Solution A) and 0.2 M NaH_2_PO_4_ H_2_O (solution B) are mixed in the appropriate amounts to obtain the desired pH 7.4. The pH is adjusted by purring solution B to A until pH = 7.4. Prepare a 0.2 M phosphate buffer comprising of 0.2 M Na_2_HPO_4_ 2H_2_O (disodium hydrogen phosphate dihydrate; for each 100 mL use 2.84 g) and 0.2 M NaH_2_PO_4_ H_2_O (sodium dihydrogen phosphate monohydrate; for each 100 mL use 2.4 g). Keep refrigerated.

Glutaraldehyde 2.5% fixative: The primary fixative for cells is 2.5% glutaraldehyde (*v*/*v*) in 0.01 M PBS.

4%. PFA: 4% PFA solution is prepared by adding 4 g of PFA in 100 mL 0.01 M PBS. Water bath is set at 60–65 °C. A beaker containing water is placed inside the water bath and 4 g of PFA is added to the PBS when it reaches a temperature of ~60–61 °C. PFA is stirred for 10–15 min but in order to dissolve completely, drops of 1 N NaOH have to be added (one drop at a time, until solution becomes transparent).

4%. Gelatin for cell pre-embedding: 4% gelatin is diluted in fresh dH_2_O according to the desired volume solution. Water bath is set at 40 °C and gelatin is added inside a beaker containing dH_2_O and stirred until completely dissolved. Before use, gelatin has to be warmed (in liquid form). 4% Gelatin can be stored at 4 °C for up to two weeks.

2%. Toluidine Blue O: 2% Toluidine blue O solution is prepared by diluting 2 g of dye in 100 mL of dH_2_O. Before use, solution must be filtered using a syringe with a 0.22 μm filter. Solution is stored in room temperature.

2%. Sodium Borate: 2% sodium borate solution is prepared by diluting 2 g of sodium borate in 100 mL dH_2_O. Before use, solution must be filtered using a syringe with a 0.22 μm filter. Solution is stored in room temperature.

Uranyl Acetate alcoholic staining solution: Uranyl acetate dihydrate has a molecular weight of 424.15 g/mol. To prepare uranyl acetate solution, we dissolve 0.375 g in 5 mL dH_2_O. Solution is stirred in a magnetic stirrer for 4 h. Once it is dissolved, 2.5 mL of 100% ethanol is added and the solution is stirred again. Solution must stay in the dark overnight to settle. Next day, the supernatant is collected and placed inside a syringe, which is then closed with parafilm. Syringe must be light-protected and stored at 4 °C.

Lead citrate staining solution: Lead citrate has a molecular weight of 1054 g/mol. To prepare lead citrate solution, we dissolve 0.04 g in 10 mL of fresh dH_2_O. Solution in stirred in a magnetic stirrer and drops of 10 N NaOH are added to the solution until it becomes transparent. Final pH has to be greater than 11. Lead citrate mustn’t stay in contact with air for long to avoid the formation of lithium carbonate precipitates in the solution. Finally, solution is placed inside a syringe which is then closed with parafilm and stored at 4 °C. Before the use, solution must be filtered using a syringe with a 0.22 μm filter.

1%. aqueous Osmium Tetroxide (OsO_4_) solution: The desired final concentration of osmium tetroxide aqueous solution (1%) is made up by dilution of the 4% stock solution. Stock and working solutions are stored at 4 °C.

Epoxy resin (Araldite): Epoxy resin can be prepared by mixing ingredients based on the following volumes.

To prepare the resin the first four ingredients are mixed inside a glass beaker very slowly with a glass stirring bar in the above ingredient order. Then DMP-30 is added in drops using a glass pipette and the mixture is stirred again slowly. Stirring has to be done slowly so as to avoid the creation of air bubbles. This resin solution will result in a standard hardness resin. Finally, resin is placed inside syringes avoiding air bubbles and are then closed up with parafilm. Resin can be stored in the freezer (~30 °C) for 3–6 months.

Acrylic resin (Lowicryl HM20): Acrylic resin is prepared according to manufacturer instructions. Specifically, the ingredients are mixed as follows:

To prepare the resin all ingredients are mixed in a glass flask using a glass stirring bar until well combined. This mixture gives blocks with medium hardness for ultraviolet polymerization. Flask is then closed with parafilm and aluminum foil. Acrylic resin has to be fresh before every use and has to be stored at 4 °C during the whole experimental procedure.

Tris buffers pH 7.2, 7.4, 7.6, 8.2: Molecular weight of Tris is 121.1 g/mol. A 0.05 M Tris buffer pH 7.6 is prepared. Afterwards, pH is adjusted for the rest of the Tris buffers by adding HCL or NaOH accordingly. Buffers can be stored at 4 °C for up two weeks.

M Glycine in Tris buffer pH 7.4: Molecular weight of glycine is 75.07 g/mol. 0.1 M glycine is diluted in Tris pH 7.4, is filtered using a syringe with a 0.22 μm filter before use and stored at 4 °C.

Blocking solution for gold immunolabelling: Solution is prepared as follows: 0.1% fish gelatin, 0.1% tween-20, 1% bovine serum albumin (BSA) and 5% normal goat serum (NGS) in Tris pH 7.4. Tris solution is mixed with tween-20 and fish gelatin using slight vortex. Afterwards, BSA is added and mixed using slight vortex. Then final solution is filtered using a syringe with a 0.22 μm and NGS is added to acquire final solution.

Primary antibody (1 ry Ab): Specific antibodies are chosen based on studied antigens. Primary antibodies are diluted in blocking solution without NGS.

Tris buffer I: Solution is prepared as follows: 0.1% tween-20 is added in Tris pH 7.4 and vortexed.

Tris buffer II: Solution is prepared as follows: 0.2% BSA and 0.1% tween-20 are added in Tris pH 7.4 and vortexed slightly.

Tris buffer III: Solution is prepared as follows: 1% BSA and 0.1% tween-20 is added in Tris pH 8.2.

BSA: BSA is diluted always in Tris pH 7.6.

Gold-conjugated secondary antibody (2 ry): 2 ry antibody is diluted in Tris buffer III. Before dilution Ab is centrifuged at 450 g for 20 min so that the supernatant is used.

### 3.3. Gold Nanoparticles

Gold nanoparticles are prepared by the Turkevich method [30], during which reduction is performed using citrates, at 100 °C. The reduction of the gold chloride solution (HAuCl_4_) is carried out with sodium citrate, bringing the gold solution to a boil. When the HAuCl_4_ solution starts to boil, the citrate solution is added. After a specified time (normally 15 min), the solution is cooled to room temperature [31].

#### 3.3.1. Citrate-Capped GNPs 15 nm Preparation

6 mg of HAuCl_4_ is dissolved in 30 mL of dH_2_O and the solution is heated to 100 °C. When boiling begins, a solution of 3 mL of dH_2_O and 30 mg of sodium citrate is added and the solution is allowed to react under mechanical agitation and heating. The reaction goes through some stages of color change. Initially, from yellow, the solution becomes colorless, then slightly blue and finally deep red. When it turns red, then the reaction stops and the liquid is allowed to come to room temperature.

#### 3.3.2. PEG-Capped GNPs Preparation

A specific amount of solution of gold nanoparticles with citrates, is purified from citrates using a semi-permeable cellulose membrane. The solution is then transferred to a vial, PEG-NH_2_ is added thereto and then placed in an ultrasonic bath. It has been observed that the particle size is affected by the concentration of PEG-NH_2_ and the residence time on ultrasound. For the synthesis of NPs close to 5 nm, the method proposed by Hussain and his team is used [32]. 4 mg of HAuCl_4_ (0.5 mM) and 8 mg PEG-NH_2_ are placed in 20 mL dH_2_O. Then, 4 mg of NaBH_4_ (50 mM) is placed in 2 mL of dH_2_O. The two solutions are then mixed and the liquid, from dark brown, turns purple.

## 4. Discussion and Conclusions

The entire experimental approach presented with TEM requires training, staff specialization and special equipment. However, the benefit in this approach is great in terms of the information obtained, since no other mean can provide it (regarding high resolution and magnification). Specific improvements and analytical steps are provided so collectively one can use TEM to follow the distribution and biological effects of NPs at the cellular level. Table 3 summarizes all the parameters regarding TEM measurements for studying GNPs radiosensitization.

More specifically, this work highlights the use of TEM as a powerful tool in the interest of studying NPs radiosensitization in correlation with ultrastructural localization of related cellular markers in vitro. TEM apart from providing morphological information on the biological structures, can be used for two main reasons: first to study the cellular uptake of different (in size, shape and coating) GNPs and second to identify specific intracellular markers in correlation with radiosensitization. For each method, we performed some elementary experiments in order to provide representative images. For this purpose, we used as an example 15 nm citrate-capped GNPs, 15 nm PEG-capped GNPs and 5 nm PEG-capped GNPs to show gold nanoparticle cellular uptake and localization. For the small and ultrasmall NPs we showed how silver-enhancement technique can be used to enlarge such particles and facilitate their detection inside the cell.

Moreover, we showed how the use of immunogold method can elucidate the ultrastructural localization of specific markers in cellular compartments. Indicatively, we used primary antibodies to detect DNA damage repair markers, such as γH2AX and OGG1, since complex DNA damage constitutes a major consequence of exposure to IR, especially when combined with radiosensitizing agents. The colocalization of DNA damage response (DDR) proteins as surrogate markers of clustered DNA damage is currently one of the most prominent ways to approach in situ this very difficult task. Fluorescence (including single molecule detection) and electron microscopy are undoubtedly the most powerful techniques in the field of radiation biology and its applications as reviewed also in Nikitaki et al. [33].

In conclusion, we aimed to describe in detail all the necessary processing steps to perform these kinds of experiments in vitro, providing also all the necessary protocols, hoping to help researchers of the same associated research field to implement TEM in their studies.

## Figures and Tables

**Figure 1 nanomaterials-11-00859-f001:**
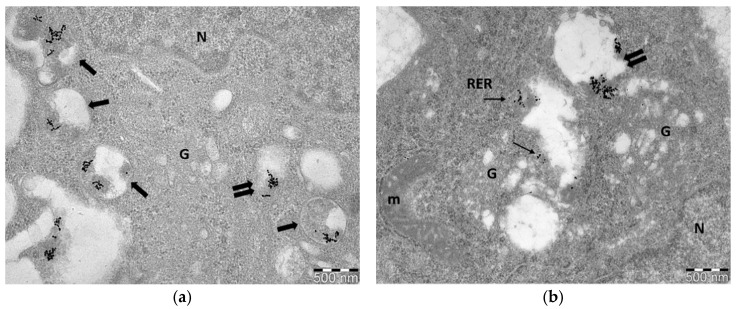
Representative TEM images of gold nanoparticle uptake in PC3 cells. (**a**) Image depicts cellular uptake of 15 nm citrate-capped GNPs; (**b**) Image depicts cellular uptake of 15 nm PEG-capped GNPs. Nanoparticles are located in vesicles (double arrows) and autophagosomes (thick arrows) but PEG-capped GNPs are also found dispersed (single or grouped particles) in the cytoplasm (thin arrows) compared to citrate-capped GNPs. N: nucleus, G: Golgi apparatus, RER: rough endoplasmic reticulum, m: mitochondrion. Fixation: 2.5% glutaraldehyde in 0.01 M PBS, embedding in epoxy resins, staining with alcoholic uranyl acetate/lead citrate. Scale bar: 500 nm.

**Figure 2 nanomaterials-11-00859-f002:**
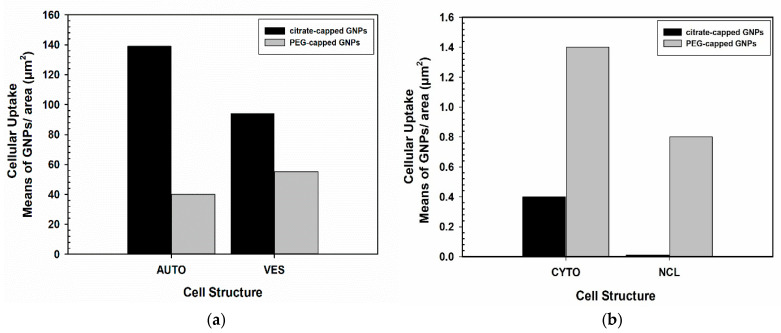
Representative data of gold nanoparticles (GNPs) uptake in PC3 cells. Data here are presented, in the form of histograms, as mean values of GNPs/area in μm^2^ of different cellular compartments: (**a**) autophagosomes (AUTO), cytoplasmic vesicles (VES); (**b**) cytoplasmic area (CYTO) and nucleus (NCL). Figure shows indicatively the cellular uptake of both citrate-capped GNPs and PEG-capped GNPs presented in Figure 1.

**Figure 3 nanomaterials-11-00859-f003:**
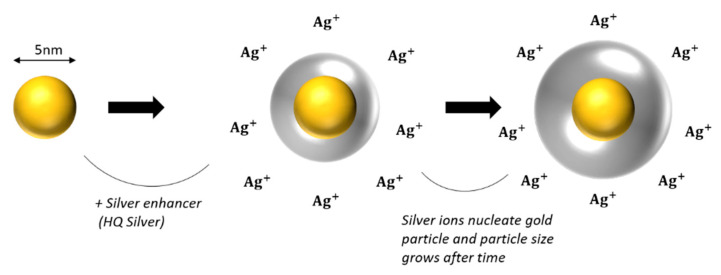
Silver-enhancement technique. Figure shows how silver nucleates gold nanoparticles, resulting in larger visible particles. Final particle size is time-dependent (increases after time).

**Figure 4 nanomaterials-11-00859-f004:**
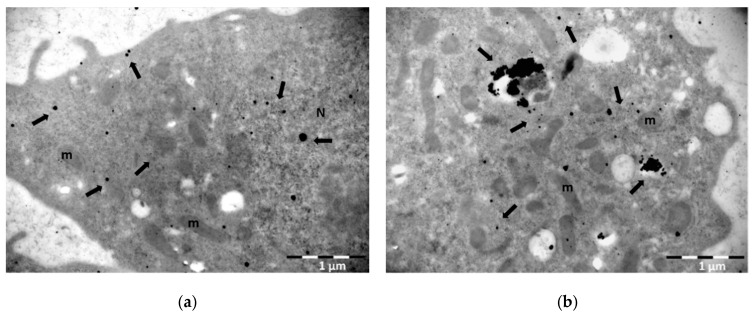
Representative TEM images of PC3 cells incubated with 5 nm PEG-capped gold nanoparticles following silver-enhancement technique: (**a**) Image shows the signal amplification after silver-enhancement of GNPs revealing their distribution both in the cytoplasm and inside the nucleus of the cell (arrows) (**b**) Part of the cytoplasm of the cell where gold nanoparticles, after silver-enhancement, are found to be dispersed or located inside vesicles and autophagosomes (arrows). N: nucleus; m: mitochondrion. Fixation: 3% paraformaldehyde and 0.5% glutaraldehyde in 0.1 M PB, embedding method: PLT, staining with alcoholic uranyl acetate/lead citrate. Scale bar: 1 μm.

**Figure 5 nanomaterials-11-00859-f005:**
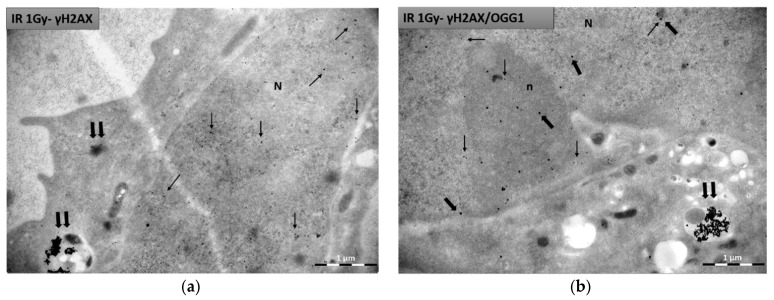
Representative TEM images of PC3 cells after immunogold labelling for DNA damage detection. 15 nm citrate-capped nanoparticles are found mostly inside cytoplasmic vesicles (double arrows), whereas 10 nm immunogold particles and 25 nm immunogold particles, labelling γH2AX and OGG1 respectively, are located inside the nucleus: (**a**) Image represents single immunolocalization for γH2Ax detection (thin arrows) inside the nucleus. (**b**) Image represents double immunolocalization for γH2Ax (10 nm gold) (thin arrows) and OGG1 (25 nm gold) (thick arrows) detection inside the nucleus. N: nucleus; n: nucleolus. Fixation: 3% paraformaldehyde and 0.5% glutaraldehyde in 0.1 M PB, embedding method: PLT, staining with alcoholic uranyl acetate/lead citrate. Scale bar: 1 μm.

**Figure 6 nanomaterials-11-00859-f006:**
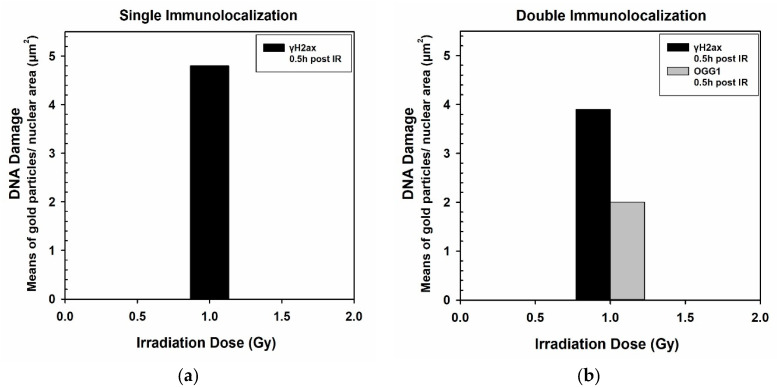
Representative data of DNA damage induced by 1 Gy IR after incubation with citrate-capped GNPs in PC3 cells. Specific markers, in this case, indicating DNA lesions, can be semi-quantified and data can be presented in the form of histograms. Figure shows indicatively the DNA damage detection by single immunolocalization of γH2AX per nuclear area (μm^2^) (**a**), or by double immunolocalization of γH2AX and OGG1 per nuclear area (μm^2^) (**b**).

**Table 1 nanomaterials-11-00859-t001:** Epoxy resin ingredient composition.

Resins	Quantity
Glycid ether 100	30.24 g
Araldite CY212	17.40 g
DDSA	52.68 g
Dibutyl Phthalate (plasticiser)	2 mL
DMP-30	74 drops

**Table 2 nanomaterials-11-00859-t002:** Lowicryl HM20 resin ingredient composition.

Resins	Quantity
Crosslinker D	7.45 g
Monomer E	42.55 g
Initiator C (powder)	0.25 g

**Table 3 nanomaterials-11-00859-t003:** Parameters for TEM Measurements/Imaging for studying GNPs radiosensitization.

Processing Steps	GNPs Cellular Uptake/Morphology	GNPs Radiosensitization through Immunocytochemistry
Fixation	2.5% glutaraldehyde solution in 0.01 M PBS	3% paraformaldehyde and 0.5% glutaraldehyde in 0.1 M PB
		PLT method/
Embedding	Epoxy resin/acrylic resin	Acrylic resin
Sectioning/grids	80 nm/copper grids	80 nm/formvar coated nickel grids
Immunolocalization	-	Single or Double immunogold labelling
Silver enhancement	YES (for small GNPs)	YES (for small gold-conjugated antibodies)
Staining	7.5% alcoholic uranyl acetate and 0.4% lead citrate	
Observation	TEM operating at 80-kV with an objective aperture of 30 μm and equipped with a digital CCD camera	
Quantification	Electron micrographs of 22,000–40,000× original magnification	
	Image analysis software (e.g., ImageJ)	

## Data Availability

Data is contained within the article or Appendix A.

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
