# Peer review of "A Guide for Using Transmission Electron Microscopy for Studying the Radiosensitizing Effects of Gold Nanoparticles In Vitro"

_nanomaterials, 2021, doi:10.3390/nano11040859_

Round 1

Reviewer 1 Report

Ioanna Tremi and coauthors present a technical note on the application of in vitro TEM studies using gold nanoparticles as radiosensitizers. The description of the working steps is extraordinarily detailed which should not be considered as native here, given it being a technical note. I consider the manuscript to give a good insight into the preparation of samples and the data analysis. However, the aspect of the TEM measurement itself is missing, which is a pity. It seems that this aspect is the last missing part to complete this contribution. I recommend that the aspect of TEM measurement, with the same level of detail, be addressed in a revision.  In summary, I am very satisfied with this manuscript and encourage its publication after a revision. In the following I would like to give a short list of small further points which should be considered.

  1. Minor typos: L.32 “3very”; 260/262 “dH20”; L.268 “fisrt”; L.549 “Dybutyl-phthalate“, just to name a few.
  2. Unit missing: L. 291 “0.02 NaOH”.
  3. 63 Please remove “especially from a specific research group“, which does not seem appropriate.
  4. Please abbreviate “nanoparticles” by NPs, whenever possible to make reading smoother.
  5. In the sections, subsections, and subsections, please use capitalization consistently or not at all.
  6. Why is the silver enhancing the biomolecular signaling and facilitate detection in TEM? Is it mainly because of the larger size or does the Ag itself promote these processes? Not clear, please clarify.
  7. The equipment listed in section 3.1. is a bit too vague at times. Please be more specific. Also I recommend to present the equipment (3.1) and materials (3.2) as paragraphs rather than a list to save space. The items of 3.1 and 3.2. should be arranged alphabetically.
  8. The aspect of colloidal stability, especially when nanomaterials are dispersed in biological media, should be discussed. Comparing PEG-coated and citrate-stabilized NPs already possess quite different colloidal stability in the presence of proteins, salts and at different pH values. The surface chemistry of NPs is decisive about their biological fate and behavior in biological environments (ACS Applied Materials & Interfaces 2015, 7, 5984.). I recommend to address the aspect of particle stabilization and to mention most commonly applied approaches to improve colloidal stability by biocompatible coatings (e.g. silica, proteins, etc.).

Author Response

RESPONSE TO REVIEWERS

We thank both reviewers and the Editor for their helpful comments, constructive criticism and suggestions. Based on all comments, we have revised the manuscript. Please see below our responses and how we address all comments point-by-point.

Reviewer 1:

Comments and Suggestions for Authors

Comment: However, the aspect of the TEM measurement itself is missing, which is a pity. It seems that this aspect is the last missing part to complete this contribution. I recommend that the aspect of TEM measurement, with the same level of detail, be addressed in a revision.

Response: We thank the reviewer for this comment. We have written some text in the manuscript regarding the technical parameters for TEM imaging reading “In order to obtain high-quality images of high resolution, Transmission Electron Microscope should operate at least at 80 kV accelerating voltage with an objective aperture of 30μm and be equipped with a digital CCD camera, such as Olympus Morada or Olympus MegaView G2, calibrated according to the manufacturer.”. We also have added certain phrases (see p.7 line 327, p.8 line 369) to emphasize on individual aspects that affect TEM imaging. We would also like to point out that regarding the aspect of TEM imaging and our experience several other parameters that play key role are described separately in each section of the manuscript. More specifically, the choice of fixative (p.4, lines 172-180), the use of OsO4 (which is omitted in PLT method) (p.5, lines 227-233), the use of formvar coated grids (p.8, lines 366-369) and the staining of thin sections (p.7, lines 325-327). Finally based on all of these parameters we also added a table in the discussion section (Table 3). We hope that our additions are adequate in order to address this comment.

(1) Minor typos: L.32 “3very”; 260/262 “dH20”; L.268 “fisrt”; L.549 “Dybutyl-phthalate”, just to name a few.

Response: All typos have been corrected throughout the manuscript.

(2) Unit missing: L. 291 “0.02 NaOH”.

Response: Missing concentration unit has been added.

(3) 63 Please remove “especially from a specific research group”, which does not seem appropriate.

Response: We have rephrased the sentence to read “However, TEM has been already used to study and quantify radiation damage (both for low- and high-LET) and more explicitly DNA damage”.

(4) Please abbreviate “nanoparticles” by NPs, whenever possible to make reading smoother.

Response: Words such as “nanoparticles” and “gold nanoparticles”, have been abbreviated, whenever was possible throughout the manuscript.

(5) In the sections, subsections, and subsections, please use capitalization consistently or not at all.

Response: The inhomogeneous use of capitalization has been corrected both in sections and subsections of the manuscript. The same applies to the supplementary material as well.

(6) Why is the silver enhancing the biomolecular signaling and facilitate detection in TEM? Is it mainly because of the larger size or does the Ag itself promote these processes? Not clear, please clarify.

Response: We thank the reviewer for this comment. Small sizes of gold nanoparticles are not easily detected through TEM, except in very high magnification. Silver enhancement technique is used to enlarge the small gold nanoparticles, as silver encapsulates them (due to surface charges) and consequently make them bigger.

In order to clarify this issue, the following paragraph on Page 10, line 454-468 in the manuscript has been rephrased and extended: “Gold NPs are visible though TEM due to their high electron density. However small sizes of NPs (<5nm) are not easily detected even in very high magnification. For that reason, silver-enhancement technique is often applied to enlarge the size of GNPs and facilitate the detection of small or ultrasmall ones. The reaction between GNPs and silver is specific for the first time period. Gold nanoparticles will nucleate the deposition of dense silver particles which will enlarge after a rapid period of time resulting in easily detectable NPs (see Figure 3). However, silver-enhancement technique is time-dependent and may give a small amount of background signal depending on the incubation time with reagent solutions. Since, after prolonged incubation with silver reagent, silver may be precipitated due to self-nucleation. In the context of experiments monitoring gold nanoparticles cellular uptake, we used 5nm colloidal gold nanoparticles capped with PEG. PEG enhances biocompatibility and reduces aggregates formation. The incubation time with silver reagents using the nanoprobes HQ silver-enhancement kit is 6min, which resulted after standardization procedure in order to acquire an adequate amplified signal, with minimum background” 

(7) The equipment listed in section 3.1. is a bit too vague at times. Please be more specific. Also, I recommend to present the equipment (3.1) and materials (3.2) as paragraphs rather than a list to save space. The items of 3.1 and 3.2. should be arranged alphabetically.

Response: We thank the reviewer for this suggestion. More information has been added accordingly: Equipment (3.1) and materials (3.2) have been corrected and changed from lists to paragraphs. The items of 3.1 and 3.2 have been arranged alphabetically.

(8) The aspect of colloidal stability, especially when nanomaterials are dispersed in biological media, should be discussed. Comparing PEG-coated and citrate-stabilized NPs already possess quite different colloidal stability in the presence of proteins, salts and at different pH values. The surface chemistry of NPs is decisive about their biological fate and behavior in biological environments (ACS Applied Materials & Interfaces 2015, 7, 5984.). I recommend to address the aspect of particle stabilization and to mention most commonly applied approaches to improve colloidal stability by biocompatible coatings (e.g., silica, proteins, etc.)

Response: We thank the reviewer for this comment and in order to provide more information regarding colloidal stability of NPs in biological media we have implemented a paragraph with the suggested reference on page 2, lines 52-74 in the manuscript reading: “In this paper we used colloidal solutions of two types of gold nanoparticles (GNPs), citrate-capped GNPs and PEG-capped GNPs. It should be noted that, colloidal stability of nanoparticles especially when they are dispersed in biological media is crucial in order to preserve their correct use as aggregation leads to different biological responses. The main factors of colloidal stability are pH and ionic strength. The physicochemical properties of the NPs are predefined by their structure and material but are also highly affected by their interaction with the environment in terms of temperature, pH, salt, proteins and different cells. Several compounds are utilized to lower the surface tension (interfacial tension) between a liquid and a solid, between two liquids or between a gas and a liquid, acting as detergents, wetting agents, emulsifiers, foaming agents, or dispersants. The surface coating chemistry since it is the first part of the nanoparticle that comes in contact with the biological environment, it does not only provide biocompatibility but also it defines their colloidal stability and their fate (uptake, excretion, degradation). Various coatings have been developed to improve the stability and biocompatibility of GNPs such as polymers (e.g., Polyethylene glycol (PEG)), thiols, citrate, peptides, lipids and other surfactants or inorganic coatings such as silica. Many of these coatings for example charged ligands (e.g., citrate) or other surfactants do not stabilize the NPs sufficiently especially in the presence of biological media, salts or protein containing media. On the other hand, polymeric coatings (e.g., PEG) provide NPs that are stable in buffer solutions or cell culture media (containing free proteins or not). Finally, protein-coated NPs also exhibit high colloidal stability in biological environments, in the presence of salt and of other proteins and are highly stable even inside biological media. It should be noted though that since proteins are sensitive to environmental pH, the stability of protein-coated NPs is also pH dependent.”.

Reviewer 2 Report

This is a technical paper describing the step needed to study the damage induced by IR and how the nanogold particle may increase and focalize the damage.

The topic of the paper is interesting and novel. The authors state that with the use of Transmission Electron Microscopy they can study the morphological alteration induced by IR and also localize them ultra-structurally. However, they attempt to describe all the procedure for conventional and immune-EM resulting in superficial description of key passage that are already well covered in books and methods paper. Ad example all the part relative to sectioning can be cut short since the personnel required to cover that task has to be skilled and trained appropriately and will not take any advantage in reading a superficial description in this paper.

I would suggest concentrating better on the part that are specific to topic of this paper. In particular there is no mention or comparison of the ultrastructural alteration of various organelles of the cell after irradiation. In particular mitochondria are not taken in consideration at all.

Conventional TEM may show the ultrastructural early alteration on mitochondria, reticulum and autophagocitic process.

It would be good to have a comparison between irradiated cells with and without nanoparticles to correctly appreciate the extent of morphological alteration.

Figure 1 refer to a conventional TEM of cells monolayer exposed to nanoparticles. The images are poorly contrasted especially regarding membranes, and the organelles are not readily detectable. I would expect the images be more informative.

Furthermore, the legend of fig 1 report that nanoparticles are inside autophagosomes. The definition of autophagosomes refer to a structure involved in degradation of intracellular content. In this case the nanoparticles are applied outside the cells and they are internalized via phagocytosis, therefore the structures containing nanoparticles are phagosomes.

The arrows used to identify different structures should be different (as used in fig 5).

The legend of TEM images should report that technique used for fixation, embedding and immunostaining (if present).

Minor: line 143 PBS in the fixative should be 0.1 M.

Author Response

RESPONSE TO REVIEWERS

We thank both reviewers and the Editor for their helpful comments, constructive criticism and suggestions. Based on all comments, we have revised the manuscript. Please see below our responses and how we address all comments point-by-point.

Reviewer 2:

Comments and Suggestions for Authors

Comment: However, they attempt to describe all the procedure for conventional and immune-EM resulting in superficial description of key passage that are already well covered in books and methods paper. Ad example all the part relative to sectioning can be cut short since the personnel required to cover that task has to be skilled and trained appropriately and will not take any advantage in reading a superficial description in this paper.

Response: We thank the reviewer for this comment. The sectioning part was cut short to avoid superficial description.  Even though the rest of the manuscript is also written in detail, we believe that all of the information is needed in order to help the reader understand the aspect of each step and be able to reproduce every process presented. Also, we would like to report that, reviewer 1 suggested that since this is a technical note every detail in the text is needed for the reader and for that reason we avoided changing and shortening the manuscript further.

Comment: In particular there is no mention or comparison of the ultrastructural alteration of various organelles of the cell after irradiation. In particular mitochondria are not taken in consideration at all.

Conventional TEM may show the ultrastructural early alteration on mitochondria, reticulum and autophagocitic process.

It would be good to have a comparison between irradiated cells with and without nanoparticles to correctly appreciate the extent of morphological alteration.

Response: We thank the reviewer for this important comment. Ultrastructural alterations of cellular compartments (such as mitochondria, cytoskeleton, lysosomes, autophagosomes and nucleus) are indeed present after ionizing irradiation (IR), and is something that must be taken into account. Since this is a technical note, we did not include information about the morphological changes after IR (with or without NPs), because in this paper we wanted to focus only on the technicalities that one has to follow in order to use TEM to study NP-induced radiosensitization. This includes identifying specific markers by immunolocalization, as well as observing the cellular uptake of GNPs. TEM many times is used only for observing ultrastructural changes. For that reason, we also wanted to present indicatively how one can also quantify the above parameters. Following this technical note, we will soon publish our research results on GNPs radiosensitization, which will address in detail the cellular uptake of GNPs, the DNA damage as well as all the cellular/subcellular morphological changes that occur after IR and after increased oxidative stress due to the use of GNPs.  However, to highlight this important aspect we have included the following paragraph concerning the morphological changes of organelles after IR on page 3, lines 90-103:” At a cellular and subcellular level radiation response may manifest in minor reversible or irreversible morphological changes, depending on radiation dose, type of radiation as well as cellular type. These changes generally refer to either the cellular shape itself or to other organelles such as mitochondria, cell membrane, autophagosomes, lysosomes, cytoskeleton, endoplasmic reticulum and Golgi complex. More precisely, changes in the number and size of microvilli, retraction of pseudopods, or rounding up of flatten cells (e.g., endothelial cells, human fibroblasts) may be observed following radiation. Concerning the cytoskeleton, disorganization of actin network and disruption of filaments have been observed in several types of cells exposed to IR, in some cases in doses ranging from 0.5Gy-1Gy. Many studies also report changes in mitochondria, such as elongation, branching and increase in size or disruption of inner and outer membranes. Golgi fragmentation or rearrangement as well as changes in the number and size of lysosomes and autophagic vacuoles are also present after exposure to IR. Due to all of these post IR effects it would be very beneficial to use electron microscopy to study these morphological changes after IR with the presence and absence of GNPs.

Comment: Figure 1 refer to a conventional TEM of cells monolayer exposed to nanoparticles. The images are poorly contrasted especially regarding membranes, and the organelles are not readily detectable. I would expect the images be more informative.

Response: Contrast has been enhanced in order to improve image quality. We also added letter m in this figure to indicate mitochondrion.

Comment: The legend of fig 1 report that nanoparticles are inside autophagosomes. The definition of autophagosomes refer to a structure involved in degradation of intracellular content. In this case the nanoparticles are applied outside the cells and they are internalized via phagocytosis, therefore the structures containing nanoparticles are phagosomes.

Response: We thank the reviewer for this comment. Nanoparticles (NPs) are indeed internalized inside phagosomes. However, after internalization NPs are also located inside other structures and more specifically autophagosomes and lysosomes. The accumulation of NPs inside autophagosomes is referenced inside the manuscript in p. 9, lines 397-399. To avoid confusion, we would also like to mention that in this manuscript with the term vesicles we refer to all vacuolar structures, except from autophagosomes and lysosomes.

Comment: The arrows used to identify different structures should be different (as used in fig 5).

Response: Different arrows have been used to identify different cellular structures.

Comment: The legend of TEM images should report that technique used for fixation, embedding and immunostaining (if present).

Response: Techniques used for fixation, embedding, immunostaining (if present), have been implemented in all figure legends.

Comment: Minor: line 143 PBS in the fixative should be 0.1 M.

Response: Regarding PBS solution the concentration is 0.01M. Regarding PB buffer the concentration is 0.1M.